# Epidemiology and clinical features of Birt-Hogg-Dubé syndrome: A nationwide population-based study in South Korea

Hyung Jun Park[1], Ye-Jee Kim[2], Min-Ju Kim[2], Ho Cheol Kim[1]*

1 Department of Pulmonary and Critical Care Medicine, Asan Medical Center, University of Ulsan College of Medicine, Seoul, South Korea, 2 Department of Clinical Epidemiology and Biostatistics, Asan Medical Center, University of Ulsan College of Medicine, Seoul, South Korea

* kimhocheol10@gmail.com

## Abstract

### Background

Birt–Hogg–Dubé (BHD) syndrome is an ultrarare lung disease with unclear prevalence and incidence. Our study aimed to identify the epidemiological and clinical features of BHD syndrome by using nationwide claims data from the Korean Health Insurance Review and Assessment service.

### Methods

Patients with BHD syndrome who had the following criteria were included: 1) tested for folliculin gene mutation, and 2) had at least one of the conditions: other specified malformation syndromes, fibrofolliculoma, acrochordon, lung cyst, cancer, and pneumothorax based on International Classification of Disease–10 code.

### Results

We found 26 patients with BHD syndrome from 2017 to 2019. The prevalence of BHD syndrome was 5.67 per $10^7$ population, with no peak age. Among incidence cases, the median age of diagnosis was 51 years, with slightly more females than males (n = 15, 57.7%). Over half of the patients (n = 14, 53.8%) experienced pneumothorax, and 10 (38.5%) developed malignant neoplasm within the clinical course.

### Conclusions

The prevalence of BHD syndrome in Korea is extremely low. However, affected patients manifest several comorbidities, including malignant neoplasm and repetitive pneumothorax.

**Data Availability Statement:** The datasets generated and/or analysed during the current study are available in the HIRA database of South Korea repository. (https://opendata.hira.or.kr, Tel: +82-

1600-2000) While HIRA restricts the data from being shared publicly, interested researchers can apply for access to the data at the HIRA web site. The researchers can request the same periods, terms and items (claim code) as done in this study.

**Funding:** The authors received no specific funding for this work.

**Competing interests:** The authors have declared that no competing interests exist.

## Introduction

Birt–Hogg–Dubé (BHD) syndrome is a rare inherited autosomal dominant disorder [1]. It is believed to be caused by the germline mutation of folliculin (FLCN) gene [2, 3]. However, the exact pathogenesis and function of FLCN remains unknown. Although FLCN mutations are often found in affected family members [4, 5], de novo development of BHD syndrome with no prior family history can also occur [6].

Typically, BHD syndrome is characterized by skin lesion, renal cancer, cystic lung disease, and spontaneous pneumothorax [7]. However, the spectrum, onset time, and frequency of these clinical manifestations are diverse, making the diagnosis difficult [8, 9]. In addition, the genetic, epidemiologic, and clinical characteristics of BHD syndrome might differ between Asian and Western populations [10, 11]. Currently, the nationwide data about its incidence, prevalence, and accompanying comorbidities are unavailable. Thus, this study aimed to investigate the epidemiologic data and clinical characteristics of BHD syndrome by using the nationwide claims data of Korea.

## Materials and methods

### Study design and data collection

This nationwide retrospective cohort study enrolled patients diagnosed with BHD who also had an ICD-10 code and an insurance payment code. In South Korea, the National Health Insurance Service is a universal insurance system that provides both inpatient and outpatient healthcare services for nearly all citizens, which enables to the investigation of the prevalence of disease in the whole national population [12, 13]. Similar to previous code-based nationwide research, we conducted an analysis based on ICD-10 codes [14, 15]. Considering its extreme rarity, the BHD syndrome does not have its own ICD-10 code; rather, it belongs to the ICD code Q878, which includes Alport syndrome, arterial tortuosity syndrome, and other congenital malformation diseases. Moreover, the confirmation test, that is, the FLCN gene test, is not common; thus, it was included in the code C5808. S1 Table lists the tests included in C5808. Patients with BHD syndrome in Korea are under the insurance policy for incurable rare diseases, which have distinct codes. However, given that BHD syndrome is rare, it is included in the rare incurable disease code V900, which consists of 127 diseases (S2 Table). Considering this limitation, the BHD syndrome was also defined using the insurance payment code for FLCN gene (C5808) and the comorbidities according to the corresponding ICD codes.

From the Health Insurance and Review Agency (HIRA) database between January 2007 and October 2019, we extracted patients undergoing FLCN gene tests using national health insurance payment code (C5808) after 2016 and subclass of gene test (C5808446) after 2017. As the presentation can differ among patients due to different disease onsets per organ, we defined BHD patients as those tested for FLCN gene mutation (C5808446) and having at least one of: other specified malformation syndromes (Q87.8), fibrofolliculoma (D23), acrochordon (L918), lung cyst (Q330), renal cancer (C64, C65, and cancer registration code (V027, V193, and V194)), and pneumothorax (J93) based on International Classification of Disease-10 (ICD-10). For distinction from other cystic lung diseases, the following diseases were excluded: Langerhans cell histiocytosis (J848) and lymphangioleiomyomatosis (D181). In BHD patients, demographic characteristics and associated diseases including malignancy were described. To identify the diseases associated with BHD syndrome, we extracted diagnoses within 1 year before and after the FLCN gene mutation test. To protect individual privacy, anonymized and de-identified information was analyzed. To quantity pneumothorax in BHD syndrome, patient-wise pattern of hospitalization was depicted throughout the entire studied period.

### Ethics approval and consent to participate

Not required, because this study only use publicly available Health Insurance and Review Agency database.

### Statistical methods

The patients' age was described using median and 25% and 75% quantiles. To understand the natural course of cancer risks, we showed the cumulative incidence using number and percentage. As the number of included patients was small, we could not evaluate group differences by usual methods [16]. The temporal relationship between the date of FLCN gene claim and pneumothorax, which is a clue for diagnosing BHD, is shown in Fig 1.

## Results

### Prevalence of BHD syndrome

Although the FLCN gene test was introduced in 2014 in South Korea, this test was first claimed in January 2016. In the database, 26 patients had BHD syndrome (5.67 per 10 million). Among them, the number and prevalence by sex were 11 and 5.67 (95% CI: 3.71–8.32) in males and 15 and 6.60 (95% CI: 3.7–10.9) in females, respectively. BHD syndrome tended to develop more in females than in males. The median (interquartile range) age was 51 (34–58) years, with 46 (30–57) years in males and 53.5 (31–59) years in females specifically. Moreover, 11 of 26 patients (42.3%) with BHD syndrome underwent FLCN gene test around 50–59 years of age and 8 of 26 (30.8%) patients were tested around 30–49 years of age. The distribution of diagnosis age according to sex is described in Table 1.

### Comorbidities

Among the 26 BHD patients, 10 patients had malignancy in this cohort observational period. Among the patients, nine (34.6%) had malignancy before the FLCN gene test and one was diagnosed with cancer after this test (Table 2). In terms of cancer subtypes, digestive cancer (15.4%) and urologic cancer (11.5%) showed the first and second highest incidence before and after BHD syndrome diagnosis, respectively. Lip and orophaynx, thoracic, genital, and hematologic cancers were also identified. Table 3 summarizes the associated diseases identified within 1 year before and after the FLCN gene test. The common comorbidities other than respiratory disease according to organ were gingivitis and periodontal disease (84%) and gastritis and duodenitis (88%). We also found gastroesophageal reflux disease (65.4%), ophthalmic disease (53.8%), dyslipidemia (53.8%), allergic contact dermatitis (42.3%), and hypertension (38.5%) (Table 3). Meanwhile, four patients (16%) had fibrofolliculoma identified within 1 year of the FLCN gene test.

### Pneumothorax

During the study period, 17 (65.4%) patients suffered at least one pneumothorax, 10 of whom (58.8%) had recurrent pneumothorax (Table 4). Pneumothorax was more common in females than in males (11, 64.7% vs. 6, 35.2%). The median age of patients at pneumothorax diagnosis was 43 (31–57) years (25%–75%), with 40 (28–57) years in males and 43 (34–55) years in females specifically. Pneumothorax occurring more than once was found in 5 of 6 male patients (83%) and in 5 of 11 female patients (45%). In those with at least one pneumothorax event, the mean (standard deviation) length of hospital stay was 9.9 (6.1) days in the first event and 12.6 (6.8) days in the second event. All the patients with recurrent pneumothorax (n = 10), no patients underwent surgery and 70% (n = 7/10) were treated with chest tube insertion. The

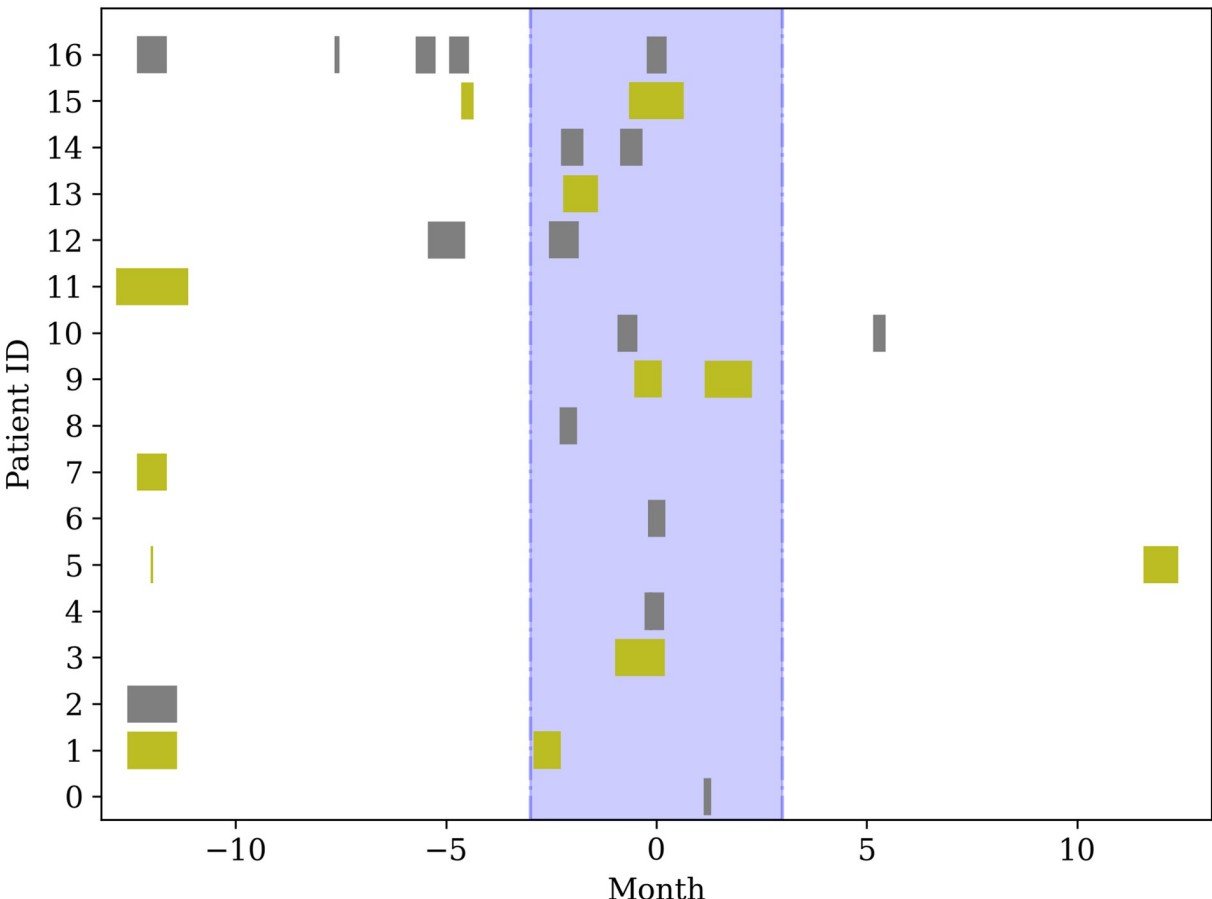

**Fig 1. Hospitalization for pneumothorax compared with date of BHD diagnosis.** The figure shows the onset and in-hospital period for pneumothorax per patient. The month zero is defined by the day of BHD diagnosis, and the shaded area is the previous 3 months of the diagnosis. The hospital admission period is depicted as the length of each bar.

**Table 1. The number of BHD patients according to age group and sex.**

| Age group | Total | | Male | | Female | |
|---|---|---|---|---|---|---|
| | N | Prevalence 95% CI | N | Prevalence 95% CI | N | Prevalence 95% CI |
| 10–19 | 1 | 2.01 (0.05–11.2) | 0 | 0 NA | 1 | 4.18 (0.11–23.3) |
| 20–29 | 3 | 4.40 (0.91–12.8) | 2 | 5.58 (0.68–20.1) | 1 | 3.09 (0.08–17.2) |
| 30–39 | 4 | 5.65 (1.54–14.4) | 2 | 5.50 (0.67–19.9) | 2 | 5.81 (0.7–21) |
| 40–49 | 4 | 4.77 (1.3–12.22) | 2 | 4.69 (0.57–16.9) | 2 | 4.85 (0.59–17.5) |
| 50–59 | 11 | 12.69 (6.34–22.7) | 4 | 9.16 (2.5–23.4) | 7 | 16.27 (6.54–33.5) |
| 60–69 | 2 | 3.16 (0.38–11.4) | 1 | 3.24 (0.08–18.0) | 1 | 3.10 (0.08–17.2) |
| 70–79 | 1 | 2.78 (0.07–15.4) | 0 | 0 NA | 1 | 5.02 (0.13–28.0) |
| total | 26 | 5.67 (3.71–8.32) | 11 | 4.76 (2.38–8.52) | 15 | 6.60 (3.7–10.9) |

NA: Not applicable; N: Number of BHD patients; Prevalence: calculated per 10 million

**Table 2. Associated malignancy in patients with BHD syndrome before and after folliculin gene test.**

| ICD-10 code | Cancer type | Before (N = 26) | After (N = 26) | Total (N = 26) |
|---|---|---|---|---|
| C00–C97 | All kinds of malignancy | 9 (34.6%) | 1 (3.8%) | 10 (38.5%) |
| C15–C26 | Digestive | 3 (11.5%) | 1 (3.8%) | 4 (15.4%) |
| C64–C68 | Ureter and bladder | 3 (11.5%) | 0 (0%) | 3 (11.5%) |
| C00– C14 | Lip and oropharynx | 2 (7.7%) | 0 (0%) | 2 (7.7%) |
| C30–C39 | Respiratory and intrathoracic | 1 (3.8%) | 0 (0%) | 1 (3.8%) |
| C51–C63 | Genital | 1 (3.8%) | 0 (0%) | 1 (3.8%) |
| C81–C96 | Hematologic | 1 (3.8%) | 0 (0%) | 1 (3.8%) |

The date of malignancy contains all periods of the patients before and after the folliculin gene mutation test.

median length of hospital stay by sex was 8 (6.5–13) in males and 9.5 (5.75–15.25) in females. Pneumothorax that occurred within 3 months of BHD syndrome diagnosis was found in 13 of all 26 patients (Fig 1). The median (interquartile range) time from pneumothorax to BHD syndrome diagnosis was 39 (5.4–98) months.

## Discussion

The current study is the first population-based research performed on the prevalence of BHD syndrome in Korea by using a nationwide claims database. Although the prevalence of BHD syndrome was extremely rare in South Korea (5.67 per 10 million), various comorbidities, including malignancy and repetitive pneumothorax, were found.

Given that the BHD syndrome is a rare inherited disorder [17], its incidence and prevalence remain unknown. Several previous reports focused on the prevalence of BHD syndrome in patients with spontaneous pneumothorax [18, 19] or family history [5, 15, 16]. Recently, Hu et al. conducted a literature review of a large BHD syndrome cohort (120 families with 221 cases) in China [20]. However, they collected information using only published data; hence, the research was not representative of the nationwide data. In South Korea, Lee et al. reported only 12 patients (10 patients confirmed by FLCN gene test) who had BHD syndrome in a single largest tertiary hospital [21], suggesting the rarity of this condition. Our current data

**Table 3. Associated diseases before and after the diagnosis of BHD syndrome.**

| ICD-10 code | Associated disease | Total (N = 26) (Number, %) |
|---|---|---|
| K29 | Gastritis and duodenitis | 23 (88.5%) |
| J00-98 | Upper respiratory disease | 23 (88.5%) |
| K02-05 | Gingivitis and periodontal disease | 22 (84.6%) |
| J20 | Acute bronchitis | 18 (69.2%) |
| K21 | Gastroesophageal reflux disease | 17 (65.4%) |
| J93 | Pneumothorax | 14 (53.8%) |
| H04, H10, H52 | Ophthalmic disease | 14 (53.8%) |
| E78 | Dyslipidemia | 14 (53.8%) |
| L03, L23, L50, M79 | Skin disease | 11 (42.3%) |
| J84 | Other interstitial lung diseases | 10 (38.5%) |
| I10 | Hypertension | 10 (38.5%) |

The date of the associated disease contains only within 1 year before and after the folliculin gene test. Upper respiratory disease includes following codes: J00, J01, J02, J03, J04, J06, J20, J30, J40, J44, and J98.

**Table 4. Number of pneumothoraxes among BHD patients.**

| Number of pneumothoraxes | Number of patients | % |
| --- | --- | --- |
| 0 | 9 | |
| 1 | 7 | 26.9% |
| 2 | 9 | 34.6% |
| >3 | 1 | 3.8% |
| Total | 26 | |

showed that BHD syndrome is ultrarare in South Korea, with a prevalence rate of only 5.67 per 10 million. As mentioned earlier, defining BHD syndrome cases using the ICD code alone seemed inadequate because it might omit other patients with BHD syndrome, leading to bias in evaluating epidemiologic data. In addition, patients with BHD syndrome who did not undergo the FLCN gene test were excluded in our analysis; this exclusion might have led to the underestimation of actual epidemiology. However, our data might serve as a reference for understanding this rare disease in South Korea.

Clinical manifestations of patients with BHD syndrome may also differ in terms of ethnicity [20, 22]. Kunogi et al. evaluated 30 Japanese patients with BHD syndrome and reported that while nearly all patients (96.7%) experienced pneumothorax, only 7 (23.3%) had skin lesion, and 2 (6.7%) had renal tumor [22]. Hu et al. investigated 221 Chinese patients with BHD syndrome and showed prevalence of pneumothorax (71.0%), skin lesion (18.1%), and renal cancer (3.6%) [20]. Thus, Asian patients might have a higher prevalence of pneumothorax and a lower incidence of cutaneous and renal manifestations than Western patients [4, 23–25]. Similarly, our current study found that pneumothorax occurred in over half of the patients (58.8%), but skin lesion (fibrofolliculoma, trichodiscoma, or accrocardon) and ureter and bladder cancer only accounted for 9 (34.6%) and 3 patients (11.5%), respectively. Recently, Liu et al. analyzed 51 Chinese patients with BHD syndrome and showed that Chinese patients had FLCN gene mutant loci that were different from those of Western patients [11]. This observation might possibly explain the ethnic difference. Notably, our current study showed several comorbidities, especially malignant neoplasm, in patients with BHD syndrome. Various tumors other than renal cancer might also occur in these patients [5, 24, 26, 27]. However, the gold standard or guidelines on how to optimally screening and manage these patients are still unavailable. Hence, further studies are needed.

Patients with BHD syndrome are predisposed to pneumothorax [1, 7]. Zbar et al. reported that the odds ratio for spontaneous pneumothorax in patients with BHD syndrome was 32 when compared with that in the general population [28]. In the current study, 65.4% of the patients experienced pneumothorax, comparable to other previous studies (approximately 24%–76%) [4, 10, 23, 29]. In addition, the median age of pneumothorax development was 43 years, which is also comparable to other previous studies with a relatively large number of patients [17, 23, 25]. Repetitive pneumothorax might also occur [23, 29]. Toro et al. showed that 101 episodes of pneumothorax (1 to 5 times in each patient) occurred in 48 patients with BHD syndrome [29]. In addition, Gupta et al. reported that the average number of pneumothorax was 3.6 in patients with BHD syndrome [23]. Our study population showed a similar recurrence rate of pneumothorax (58.8%). Notably, the hospital stay was longer in the second episode than in the first episode, suggesting that treatment might be difficult in patients with repetitive pneumothorax. Moreover, pneumothorax within 3 months of BHD syndrome diagnosis occurred in 13 patients (50% of study population), consistent with the previous study of

Gupta et al. [23], which showed that pneumothorax was the presenting manifestation of BHD in 65% of their patients.

This study has some limitations. Considering that it used a nationwide medical claims database, BHD syndrome without FLCN gene mutation is not included in our dataset, thereby possibly omitting approximately 10% of patients with BHD syndrome [30]. Our study also did not include those with a family history of BHD syndrome and FLCN gene mutation, thereby reducing the representation of genetic features. Furthermore, if the clinician did not perform FLCN gene test for patients with suspected BHD syndrome, our criteria would underestimate the real prevalence. However, this study collected all FLCN gene tests performed in South Korea from 2016 to 2019, but only 26 patients were found. Moreover, we collected a maximum of 12.8-year follow-up data of patients suspected with BHD syndrome, including their clinical phenotypes combined with comorbidities and pneumothorax history, thereby showing the characteristics and clinical course of BHD syndrome.

## Conclusions

In conclusion, the prevalence of BHD syndrome in South Korea is low. The patients with BHD syndrome are characterized by several comorbidities, including malignant neoplasm and repetitive pneumothorax.

## Supporting information

**S1 Table. Tests included in C5808 codes.**
(DOCX)

**S2 Table. Diseases included in rare incurable disease code.** NA: not applicable due to absence of a defined code.
(DOCX)

## Author Contributions

**Conceptualization:** Hyung Jun Park, Ho Cheol Kim.

**Data curation:** Hyung Jun Park, Ye-Jee Kim, Min-Ju Kim, Ho Cheol Kim.

**Formal analysis:** Hyung Jun Park, Ye-Jee Kim, Min-Ju Kim.

**Methodology:** Hyung Jun Park, Ye-Jee Kim, Min-Ju Kim, Ho Cheol Kim.

**Supervision:** Ho Cheol Kim.

**Visualization:** Hyung Jun Park.

**Writing – original draft:** Hyung Jun Park, Ho Cheol Kim.

**Writing – review & editing:** Hyung Jun Park, Ho Cheol Kim.

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
