## [Decision Letter · Decision Letter 0]

24 Feb 2022

PONE-D-22-00655Epidemiology and Clinical Features of Birt-Hogg-Dubé Syndrome: A Nationwide Population-Based Study in South KoreaPLOS ONE

Dear Dr. Kim,

Thank you for submitting your manuscript to PLOS ONE. After careful consideration, we feel that it has merit but does not fully meet PLOS ONE’s publication criteria as it currently stands. Therefore, we invite you to submit a revised version of the manuscript that addresses the points raised during the review process.

We look forward to receiving your revised manuscript.

Kind regards,

Dong Keon Yon, MD, FACAAI

Academic Editor

PLOS ONE

Journal Requirements:

 [Unfunded studies]. 

[No authors have competing interests]. 

6. Please remove your figures from within your manuscript file, leaving only the individual TIFF/EPS image files, uploaded separately.  These will be automatically included in the reviewers’ PDF.

Additional Editor Comments:

Thank you for submitting your manuscript to Plos One. The reviewers and I believe it is of potential value for our readers. However, the reviewers have raised a number of very important issues, and their excellent comments will need to be adequately addressed in a revision before the acceptability of your manuscript for publication in the Journal can be determined.

#1. Please discuss and cite the papers below, in Page 3; Line 18-19.

A. Woo A, Lee SW, Koh HY, Kim MA, Han MY, Yon DK. Incidence of cancer after asthma development: 2 independent population-based cohort studies. J Allergy Clin Immunol. 2021 Jan;147(1):135-143. doi: 10.1016/j.jaci.2020.04.041. Epub 2020 May 15. PMID: 32417133.

B. Yoo IK, Marshall DC, Cho JY, Yoo HW, Lee SW. N-Nitrosodimethylamine-contaminated ranitidine and risk of cancer in South Korea: a nationwide cohort study. Life Cycle 2021;1:e1. https://doi.org/10.54724/lc.2021.e1

Reviewers' comments:

Reviewer's Responses to Questions

**Comments to the Author**

1. Is the manuscript technically sound, and do the data support the conclusions?

Reviewer #1: Yes

Reviewer #2: Yes

2. Has the statistical analysis been performed appropriately and rigorously? 

Reviewer #1: Yes

Reviewer #2: Yes

3. Have the authors made all data underlying the findings in their manuscript fully available?

Reviewer #1: Yes

Reviewer #2: Yes

4. Is the manuscript presented in an intelligible fashion and written in standard English?

Reviewer #1: Yes

Reviewer #2: Yes

5. Review Comments to the Author

Reviewer #1: The authors aimed to investigate the epidemiologic data and clinical characteristics of Birt-Hogg-Dube syndrome of Korea. The most important limitation of the study is that, apart from the 26 cases detected in the study, the number of BHD syndrome cases in which the FLCN mutation is not studied cannot be determined, and this was stated by the authors in the limitations section. Except that the article is well designed and written.

Reviewer #2: Wonderful study especially because of it’s rarity.Just out of my own interest,would like to ask a couple of questions.

1.What was the X ray and HRCT pattern of these patients.

2.The study said around 58% of the patients had recurrent pneumothorax.How were they managed.

6. PLOS authors have the option to publish the peer review history of their article (what does this mean?). If published, this will include your full peer review and any attached files.

Reviewer #1: No

Reviewer #2: No

---

## [Author Response · Author response to Decision Letter 0]

24 Apr 2022

March 29th, 2022

Dong Keon Yon, MD, FACAAI

Academic Editor

PLOS ONE

Manuscript number: PONE-D-22-00655

Manuscript title: Epidemiology and Clinical Features of Birt-Hogg-Dubé Syndrome: A Nationwide Population-Based Study in South Korea

Dear Dong Keon Yon 

We would like to thank you for the letter dated 24/02/2022, and the opportunity to resubmit a revised copy of this manuscript. We would also like to take this opportunity to express our thanks to the reviewers for the positive feedback and helpful comments for correction or modification. We have made every attempt to fully address these comments in the revised manuscript. Reviewer’s original comments are listed below, followed by our response to each comment. Changes made in the manuscript are marked in red. We very much hope the revised manuscript is accepted for publication in PLOS ONE.

The authors have declared that no competing interests exist. And the authors received no specific funding for this work. For data availability, HIRA (Health Insurance Review and Assessment Service) only provides summary data for researchers with limited permission to use the data. Therefore, we cannot share the minimal data set for this study. And we added the references which are well-performed studies using HIRA dataset which enable the national prevalence trend investigations.

Sincerely,

Ho Cheol Kim, MD, PhD

Department of Pulmonary and Critical Care Medicine, 

Asan Medical Center, 

88, Olympic-ro 43-gil, Songpa-gu, Seoul, 05505, South Korea 

Phone No: +82 02-3010-1494

Fax No: +82 02-3010-3130

Email Address: kimhocheol10@gmail.com

Response to Reviewers

Additional Editor Comments:

#1. Please discuss and cite the papers below, in Page 3; Line 18-19.

A. Woo A, Lee SW, Koh HY, Kim MA, Han MY, Yon DK. Incidence of cancer after asthma development: 2 independent population-based cohort studies. J Allergy Clin Immunol. 2021 Jan;147(1):135-143. doi: 10.1016/j.jaci.2020.04.041. Epub 2020 May 15. PMID: 32417133.

B. Yoo IK, Marshall DC, Cho JY, Yoo HW, Lee SW. N-Nitrosodimethylamine-contaminated ranitidine and risk of cancer in South Korea: a nationwide cohort study. Life Cycle 2021;1:e1. https://doi.org/10.54724/lc.2021.e1

- Answer: We have added these two papers as references in page 3, line 18-21.

In South Korea, the National Health Insurance Service is a universal insurance system that provides both inpatient and outpatient healthcare services for nearly all citizens, which enables to the investigation of the prevalence of disease in the whole national population.[12,13]

Reviewer #1: The authors aimed to investigate the epidemiologic data and clinical characteristics of Birt-Hogg-Dube syndrome of Korea. The most important limitation of the study is that, apart from the 26 cases detected in the study, the number of BHD syndrome cases in which the FLCN mutation is not studied cannot be determined, and this was stated by the authors in the limitations section. Except that the article is well designed and written.

- Answer: Thank you for your comment. This study investigates the prevalence of BHD syndrome and the associating malignancy and diseases. As the BHD syndrome could be diagnosed without the FLCN gene test, we agree that your point for omitting the BHD syndrome patients without FLCN gene as we discussed in limitation section. However, we thought that this method could reflect the malignancy and associating diseases of BHD syndrome even though the limitations, which help clinicians to consider the natural history of this syndrome. 

Reviewer #2: Wonderful study especially because of it’s rarity. Just out of my own interest, would like to ask a couple of questions.

1. What was the X ray and HRCT pattern of these patients. 

- Unfortunately, since we used the claim data called HIRA(Health Insurance Review & Assessment) data in our current study (ref: J Lipid Atheroscler. 2022;11:e1. Understanding and Utilizing Claim Data from the Korean National Health Insurance Service (NHIS) and Health Insurance Review & Assessment (HIRA) Database for Research, by Kyoung et al.) we could not get information about the test results. 

2. The study said around 58% of the patients had recurrent pneumothorax. How were they managed. 

- Thanks for your comments. When we re-evaluated the patient with recurrent pneumothorax (n = 10), no patients underwent surgery and 70% (n = 7/10) were treated with chest tube insertion. We added this in the result section.

---

## [Decision Letter · Decision Letter 1]

20 May 2022

Epidemiology and Clinical Features of Birt-Hogg-Dubé Syndrome: A Nationwide Population-Based Study in South Korea

PONE-D-22-00655R1

Dear Dr. Kim,

We’re pleased to inform you that your manuscript has been judged scientifically suitable for publication and will be formally accepted for publication once it meets all outstanding technical requirements.

Kind regards,

Dong Keon Yon, MD, FACAAI

Academic Editor

PLOS ONE

Additional Editor Comments (optional):

I congrature the authors on this mesmerizing paper.

Reviewers' comments:

Reviewer's Responses to Questions

**Comments to the Author**

1. If the authors have adequately addressed your comments raised in a previous round of review and you feel that this manuscript is now acceptable for publication, you may indicate that here to bypass the “Comments to the Author” section, enter your conflict of interest statement in the “Confidential to Editor” section, and submit your "Accept" recommendation.

Reviewer #1: All comments have been addressed

Reviewer #2: All comments have been addressed

2. Is the manuscript technically sound, and do the data support the conclusions?

Reviewer #1: Yes

Reviewer #2: Yes

3. Has the statistical analysis been performed appropriately and rigorously? 

Reviewer #1: Yes

Reviewer #2: Yes

4. Have the authors made all data underlying the findings in their manuscript fully available?

Reviewer #1: Yes

Reviewer #2: Yes

5. Is the manuscript presented in an intelligible fashion and written in standard English?

Reviewer #1: Yes

Reviewer #2: Yes

6. Review Comments to the Author

Reviewer #1: (No Response)

Reviewer #2: All issues have been adequately addressed.Hence,the article may be accepted for publication according to me.

7. PLOS authors have the option to publish the peer review history of their article (what does this mean?). If published, this will include your full peer review and any attached files.

Reviewer #1: No

Reviewer #2: No

---

## [Editor Report · Acceptance letter]

26 May 2022

PONE-D-22-00655R1 

Epidemiology and Clinical Features of Birt-Hogg-Dubé Syndrome: A Nationwide Population-Based Study in South Korea 

Dear Dr. Kim:

I'm pleased to inform you that your manuscript has been deemed suitable for publication in PLOS ONE. Congratulations! Your manuscript is now with our production department. 

Kind regards, 

on behalf of

Dr. Dong Keon Yon 

Academic Editor

PLOS ONE